# Extending density functional theory with near chemical accuracy beyond pure water

Suhwan Song[1,2], Stefan Vuckovic [3,4], Youngsam Kim[1], Hayoung Yu [1], Eunji Sim [1] ✉ & Kieron Burke [2,5]

Density functional simulations of condensed phase water are typically inaccurate, due to the inaccuracies of approximate functionals. A recent breakthrough showed that the SCAN approximation can yield chemical accuracy for pure water in all its phases, but only when its density is corrected. This is a crucial step toward first-principles biosimulations. However, weak dispersion forces are ubiquitous and play a key role in noncovalent interactions among biomolecules, but are not included in the new approach. Moreover, naïve inclusion of dispersion in HF-SCAN ruins its high accuracy for pure water. Here we show that systematic application of the principles of density-corrected DFT yields a functional (HF-r²SCAN-DC4) which recovers and not only improves over HF-SCAN for pure water, but also captures vital noncovalent interactions in biomolecules, making it suitable for simulations of solutions.

The properties of water, such as the uniqueness of its phase diagram, never stop surprising scientific communities.[1] Given the vital importance of water in fields that vary from material science to biology, there has been a recent surge in the development and competition of different electronic structure methods for simulating water.[2–9] As ab initio quantum-chemical methods are too expensive for large systems, Kohn-Sham density functional theory (KS-DFT) has become a workhorse of electronic structure methods for running water calculations.[10–14] But, despite an excellent accuracy to cost ratio, historically KS-DFT has been unable to deliver sufficiently high accuracy in water simulations to reproduce experimental data.[15–18]

A recent breakthrough in this direction by Dasgupta et al. showed that the strongly constrained and appropriately normed (SCAN) functional, when used in tandem with density-corrected DFT (DC-DFT), is a game changer for water simulation, because it brings KS-DFT close to chemical accuracy.[3,4] The role of water in a chemical or biochemical reaction goes beyond providing an environment to help a reaction in an aqueous solvation and is often explicitly involved in the mechanism. For this reason, a complete understanding of the reaction is possible only when the interaction between water and other molecules is accurately described. Figure 1 shows how an integratively

designed DC-DFT procedure, HF-r²SCAN-DC4, describes not only the interactions between water-water, water-organic molecules, and water-biochemical molecules in various situations, but also the interactions of noncovalent complexes at chemical accuracy or better.

DC-DFT is a general framework that separates errors of any self-consistent DFT calculations into a contribution coming from the approximate D (density) and the true error coming from the approximate F (functional).[19–22] In addition to being a rigorous exact theory, DC-DFT gives practical guidance on when and how it can be used to reduce errors in DFT simulation.[23–26] Standard DFT calculations are performed self-consistently (SC). The simplest form of practical DC-DFT is HF-DFT, where density functionals are evaluated instead on Hartree-Fock (HF) densities and orbitals.[27–31] While in most cases, SC-DFT gives the best answer, in some errors in specific cases SC-DFT suffers from large energetic errors due to the approximate density (density-driven errors).[19,22] In such cases, HF-DFT typically yields significant improvements over SC-DFT, and these include a number of chemical domains (barrier heights, some torsional barriers, halogen bonds, anions, etc.).[26]

SCAN is a non-empirical meta-generalized gradient approximation (meta-GGA) functional designed to satisfy 17 exact physical

[1]Department of Chemistry, Yonsei University, 50 Yonsei-ro Seodaemun-gu, Seoul 03722, Korea. [2]Department of Chemistry, University of California, Irvine, CA 92697, USA. [3]Institute for Microelectronics and Microsystems (CNR-IMM), Via Monteroni, Campus Unisalento, 73100 Lecce, Italy. [4]Departments of Chemistry & Pharmaceutical Sciences and Amsterdam Institute of Molecular and Life Sciences (AIMMS), Faculty of Science, Vrije Universiteit, De Boelelaan 1083, 1081HV Amsterdam, The Netherlands. [5]Departments of Physics & Astronomy, University of California, Irvine, CA 92697, USA. ✉e-mail: esim@yonsei.ac.kr

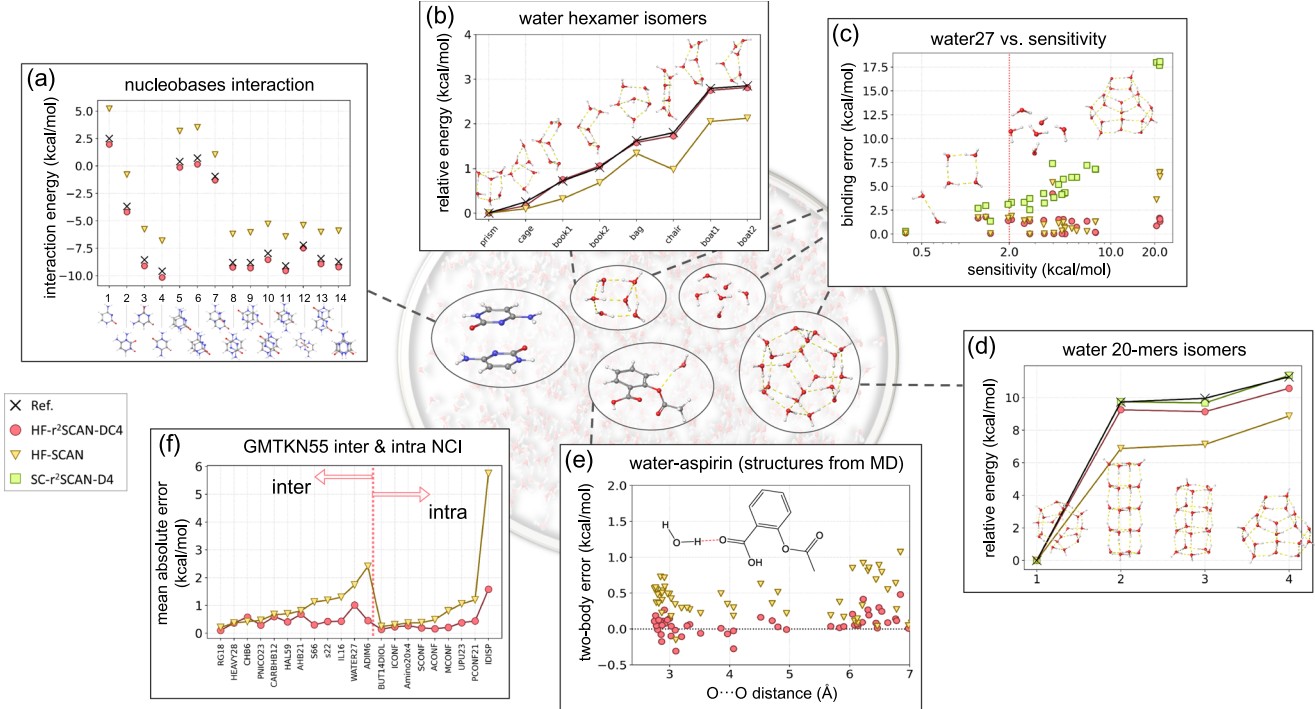

**Fig. 1 | Performance of HF-r²SCAN-DC4 relative to HF-SCAN for various chemical reactions.** Atom color code: C, gray; O, red; N, blue; and H, white. **a** the interaction energy of various configurations of the stacked cytosine dimer, where HF-SCAN underbinds by 2–3 kcal/mol; **b** energies of water hexamer relative to the lowest-lying prism isomer, with HF-SCAN underestimating by up to 1 kcal/mol; **c** errors in binding energy of WATER27 complexes as a function of density sensitivity (how much a DFT energy changes when the density is changed), showing how large errors can be without using the HF density. One cluster, $H_3O^+(H_2O)_6OH^-$ (at $x$ close to 4 kcal/mol) is an outlier argued to exhibit a significant multiconfigurational character[4]; **d** relative energies of water 20-mer isomers (not density sensitive) from WATER27, where self-consistent SC-r²SCAN-D4 performs best, but using the HF density introduces little error; **e** errors in interaction energies in the water ⋯ aspirin dimer structures from an MD simulation at $T = 298.15$ K; **f** mean-absolute-errors (MAEs) for intra- and inter-molecular noncovalent interactions datasets from the GMTKN55 database. For more details, see the main text and supplementary information.

constraints, and to recover several nonbonded norms.[32] Meta-GGA's use the KS kinetic energy density as an ingredient, but are not hybrid functionals like B3LYP[33–36], which include some fraction of exact exchange from a HF calculation.[35] In terms of accuracy, SCAN is often on par with highly empirical more expensive density functionals designed for molecules. At the same time, it enjoys great successes for simulations of extended systems, making it one of the most-used general-purpose functionals developed over the last 10 years.[37–41]

Earlier works have shown that standard (SC) DFT calculations of water clusters suffer badly from density-driven errors, which explains why HF-SCAN is much more accurate than its SC counterpart for simulations of water.[3,25] In addition to water clusters, Dasgupta et al. used HF-SCAN in tandem with many-body potential energy function related to the highly popular MB-pol[10–12] to run molecular dynamics (MD) simulations of liquid water and obtained results in excellent agreement with the experimental data. These were the first successful DFT-based simulations able to correctly describe the condensation of water.

Nevertheless, the convergence of the SCAN functional can be painfully slow with respect to the size of molecular grids, due to either the size of a system or because it would require grids larger than those available in most of the standard-quantum chemical codes.[31] Larger grids also lead to longer computational times. To address these issues of SCAN, Perdew and co-workers developed the regularized-restored SCAN functional (r²SCAN), which regularizes SCAN but restores SCAN's adherence to exact constraints.[42] But, as we show below, a standalone version of HF-r²SCAN is much less accurate for water simulations than HF-SCAN.

Despite enormous success in modelling water, HF-SCAN is not a panacea. In their water simulations, Dasgupta et al. used HF-SCAN without dispersion correction, as they found that the standard dispersion corrections, such as those of Grimme[43], worsen the original results of HF-SCAN for water. But such dispersion corrections have long been known to be necessary for noncovalent interactions (NCIs).[44–53] So, despite delivering a high accuracy for pure water simulations, HF-SCAN without a dispersion correction cannot describe accurately long-range dispersion interactions. For this reason, the errors of HF-SCAN are several times larger than those of DFT enhanced by a dispersion correction for the standard noncovalent datasets.[40] The challenge is then to construct an efficient density functional that correctly describes NCIs of different nature, while recovering or even improving the accuracy of HF-SCAN for water simulations.

In the present paper, we resolve these issues by using the principles of DC-DFT to carefully parameterize a dispersion correction for HF-r²SCAN. This yields HF-r²SCAN-DC4, which produces the following key results: (i) HF-r²SCAN-DC4 improves upon HF-SCAN for pure water simulations, by up to 0.7 kcal/mol for relative energies of water hexamers, and up to 2.4 kcal/mol for those of water 20-mers; (ii) HF-r²SCAN-DC4 is far more accurate than HF-SCAN for interactions of water with other molecules and for NCIs in general, because of the inclusion of explicit dispersion corrections; (iii) HF-r²SCAN-DC4 can be routinely and efficiently used in calculations because, unlike HF-SCAN[31], HF-r²SCAN-DC4 has no grid convergence issues. In our HF-r²SCAN-DC4, each of the three ingredients is vitally important: The HF part reduces density-driven errors, while r²SCAN fixes the grid issues of SCAN. But most importantly, the way in which we parametrize the D4 corrections by using the DC-DFT principles is vital, as an unwitting fitting of D4 ruins the accuracy for water simulations. If we drop any of those elements of HF-r²SCAN-DC4, at least one of its three appealing results will be lost.

## Results

To illustrate all these points, and how they work together, we created Fig. 1. We show how HF-r²SCAN-DC4 is better than HF-SCAN for interactions of nucleobases [panel (a)], water molecules with one another [panels (b), (c), (d)], water with other molecules [panel (e)], and NCIs in general [panel (f)].

Stacking interactions in nucleobases are of vital importance in biology as their energetics is essential to describe the formation and stability of DNA and RNA.[54,55] In Fig. 1a, we compare the accuracy of HF-SCAN and HF-r²SCAN-DC4 for interaction energies of stacked cytosine dimers at different configurations. As we can see from Fig. 1a, our HF-r²SCAN-DC4 essentially greatly reduces the errors of HF-SCAN that systematically underbinds these stacked complexes by about 2.5 kcal/mol. This demonstrates that despite its success for modeling water, HF-SCAN misses most of dispersion and thus cannot compete with our HF-r²SCAN-DC4 in modelling NCIs. This is especially the case for NCIs dominated by dispersion interactions as those present in stacked nucleobases. (See Supplementary Fig. 3 for the errors in interaction energies.) We note that the mean absolute error (MAE) of HF-r²SCAN-DC4 (0.4 kcal/mol) is very good relative to HF-SCAN, but not very impressive relative to B3LYP-D3(BJ) (<0.2 kcal/mol).[55] But such functionals include only a fraction of HF exchange, and so still suffer from large density-driven errors in water, and so have larger errors for pure water (as shown below).

Water hexamers, the smallest drops of water[56,57], are important, as they represent the transition from two-dimensional to three-dimensional hydrogen-bonding networks.[58-60] The energy differences between two adjacent isomers of water hexamers are tiny, making even the ordering of isomers a very challenging test for quantum-chemical methods.[58,61] In Fig. 1b, we compare the energies of water hexamer isomers relative to the energy of the prism, as the lowest-lying isomer.[58,62,63] Despite being more accurate for water hexamers than most DFT methods available on the market, HF-SCAN mistakes the ordering of the isomers, as it predicts too low energies of the chair isomer. Our HF-r²SCAN-DC4 is also here superior to HF-SCAN, as it not only gives the right ordering of isomers, but essentially reproduces the reference values for the relative energies of isomers. If D4 is fitted by not accounting for the DC-DFT principles (see below), the accuracy of HF-r²SCAN-DC4 for the water simulation is lost. This happened in ref. [64] and will be discussed in Methods.

We use the WATER27 dataset to illustrate the importance (and subtlety) of DC-DFT for water simulations. WATER27 is a standard dataset for binding energies of water clusters. Density sensitivity, $\tilde{S}$, is a measure for how sensitive a given DFT simulation is to errors in densities (see Supplementary Note 2 for further details and specific definitions).[23] Typically, the errors of SC-DFT calculations grow with $\tilde{S}$, indicating the presence of large density-driven errors.[26,31,65] DC-DFT reduces these large density-driven errors of SC-DFT and thus the errors of DC-DFT do not grow with $\tilde{S}$. In Fig. 1c we plot WATER27 errors as a function of density sensitivity. As the errors of SC-r²SCAN-D4 grow with $\tilde{S}$, so also does the energetic improvement of HF-r²SCAN-DC4 over SC-r²SCAN-D4. Furthermore, sometimes dispersion corrections worsen SC-DFT for cases with large density-driven errors.[25,26] This is also the case here, as SC-r²SCAN-D4 significantly deteriorates the accuracy of SC-r²SCAN (see Supplementary Fig. 4). The errors of HF-SCAN are also substantially lower than those of SC-r²SCAN-D4, and for most of the binding energies of the WATER27 clusters, HF-SCAN is comparable to HF-r²SCAN-DC4. But, for the four clusters with the largest sensitivities, HF-r²SCAN-DC4 outperforms HF-SCAN by ~4 kcal/mol.

WATER27 is a part of the GMTKN55[40], a database that we use to train the D4 parameters in HF-r²SCAN-DC4 (see Methods). But, according to the principles of DC-DFT, we exclude those WATER27 clusters that are density-sensitive, as their energetic errors are dominated by the errors in their densities.[26] Thus none of the clusters that

are to the right of the vertical dashed line placed at $\tilde{S} = 2$ kcal/mol (see Methods for the details on this reasoning) are used in the fitting, which means HF-r²SCAN-DC4 makes genuinely accurate predications for a vast majority of these water clusters. Not only does it recover HF-SCAN for binding energies of the water clusters, but also provides substantial improvements for the most challenging clusters.

An important question is whether or not one should always correct the density. The general principles of DC-DFT say that one should only correct the density in cases of substantial density-driven errors. In density insensitive cases, the effect of correcting the density should be small, and may actually worsen energetics. Figure 1d shows energies of water 20-mers relative to the energy of the lowest of the four 20-mers. Here SC-r²SCAN-D4 beats its DC counterpart every time. In contrast to large $\tilde{S}$ for binding energies of the four 20-mers (the last four data-points in Fig. 1c), the sensitivities corresponding to their relative isomer energies are about twenty times smaller (Supplementary Fig. 7). Thus the higher accuracy of SC-r²SCAN-D4 over HF-r²SCAN-DC4 does not come as a surprise. But the crucial point is that, even in this low-sensitivity scenario, the errors introduced by the HF density are far smaller than those of HF-SCAN, and remain tiny on a per molecule basis.

A crucial figure of merit is how accurate energetics are for water molecules in the vicinity of an organic molecule, especially if it is polar. In Fig. 1e, we show errors in the interaction energies between water and aspirin from structures that we extracted from an MD simulation at $T = 298.15$ K (see Supplementary Note 6 for further details on the MD simulation). The structures are sorted by the distance between the oxygen atom in water and the specified oxygen atom in the carboxyl group of aspirin. The errors of HF-r²SCAN-DC4 are much smaller than those of HF-SCAN. They are also substantially smaller than those of SC-r²SCAN-D4 (Supplementary Fig. 8), demonstrating again the importance of both the D4 and DC components in our method.

Getting NCI right across a broad range of molecules is important, even in the absence of water. The GMTKN55 collection of 55 databases has become a standard benchmark[40] and includes many databases for NCIs. In Fig. 1f, we compare the MAEs of HF-SCAN and HF-r²SCAN-DC4 for the standard datasets with intra- and intermolecular NCIs.[40] Despite its high accuracy for water clusters, HF-SCAN does not capture long-ranged dispersion interactions. This is why it is far less accurate than HF-r²SCAN-DC4 for noncovalent datasets. We can see that HF-r²SCAN-DC4 is highly accurate here, and on average it beats SC-r²SCAN-D4 for both inter- and intramolecular NCIs (see Supplementary Table 1 comparing the metrics for overall performance).

### Interaction energies for water dimers

As discussed already, HF-SCAN performs incredibly well for interactions in pure water. In this section, we look at selected water dimers that are relevant to water simulations, and show how HF-r²SCAN-DC4 reproduces (or even exceeds) this accuracy. More importantly, we show how each aspect of its construction (density correction, regularization of SCAN, and dispersion correction) is vital to its accuracy for water. Later we will show that no other approximation at this level of cost comes close to this performance for water.

Figure 2 shows the interaction energies for many water dimers (the difference in the energies of a dimer and two monomers). (a) shows the interaction energies at Smith stationary points, some of which resemble geometries from dense ice structures.[18] (b) shows the errors of approximations in interaction energies for water dimers as a function of the distance between the two oxygen atoms. The underlying structures were extracted from an MD simulation at $T = 298.15$ K (see Supplementary Note 6 for further details on the simulation). For the interaction energies of these water dimers, HF-SCAN without a dispersion correction already provides a very high accuracy (with MAEs of <0.1 kcal/mol). Our HF-r²SCAN-DC4 essentially recovers this

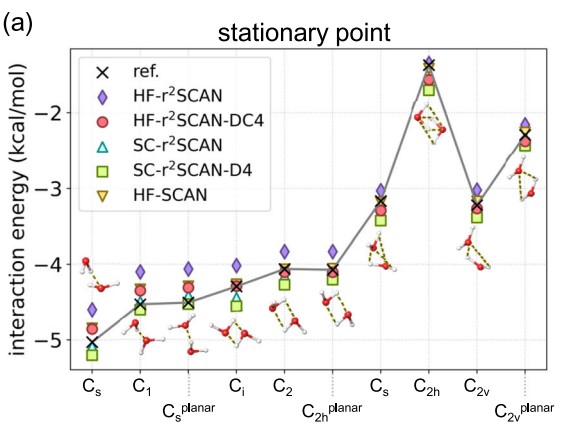

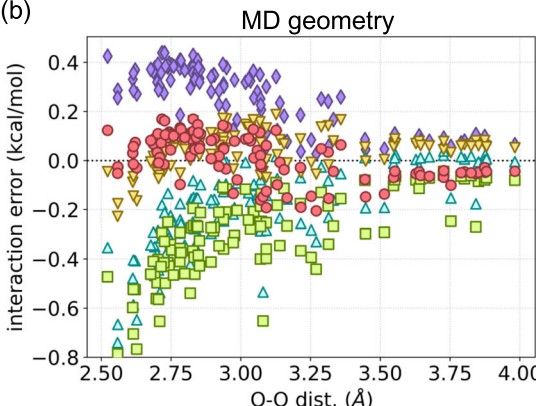

**Fig. 2 | Water dimer interaction energies. a** Smith stationary points[73] and **b** MD simulated water dimers with the oxygen-oxygen distance. For **a**, MAEs of each functional are (following the order in the legend) 0.25, 0.11, 0.09, 0.17, and 0.08 kcal/mol. DLPNO-CCSD(T)-F12 has been used as a reference. For **b**, MAE of each functionals are 0.25, 0.08, 0.20, 0.30 and 0.08 kcal/mol. Supplementary Fig. 1 shows the corresponding density sensitivities and Supplementary Fig. 2 shows the errors of approximations for the Smith dimers and interaction energies for MD dimers.

high accuracy of HF-SCAN. Similar patterns observed for binding energies of water clusters are also seen here.

By studying the various plots, one can assess the importance of the relative contributions to HF-r$^2$SCAN-DC4. First, the purple points give HF-r$^2$SCAN, to be contrasted with HF-SCAN. We see that HF-r$^2$SCAN significantly (on this scale) underestimates the interaction energy. Even though r$^2$SCAN was designed to reproduce the results of SCAN, these differences are so small as to be negligble for most purposes. However, they are clearly significant here, showing HF-r$^2$SCAN is noticeably less accurate for these dimers. The addition of the D4 correction, however, makes their errors comparable.

On the other hand, we may also consider the importance of density correction. We see that SC-r$^2$SCAN-D4 considerably overestimates interaction energies. In fact, SC-r$^2$SCAN does rather well, as the errors due to poor density and missing dispersion cancel.

We can also observe from Fig. 2b that the improvement of HF-r$^2$SCAN-DC4 over SC-r$^2$SCAN-D4 decreases with the distance between the two oxygen atoms in water dimers. This can be understood in terms of underlying density sensitivity which also decreases with the O-O distance (see Supplementary Fig. 1).

**Many-body interactions in larger water clusters**
In Fig. 3 we compare errors of HF-r$^2$SCAN-DC4 and HF-SCAN for the interaction energies of the eight standard water hexamers.[58,59] In addition to total interaction energies, we also use the many-body expansion (MBE) to show the $K$-body contributions to these energies (with $K$ in between 2 and 6). This is a standard methodology for understanding the origins of errors in water models.[3,13,66] The energetic importance of the $K$-body contributions decreases rapidly with $K$ (Supplementary Fig. 6), making the 2-body contributions by far the most important, and these are where significant differences emerge when the density is corrected. But in order to reach chemical accuracy, a proper description of the higher-order contributions also matters. The 2-body plot shows that HF-SCAN has a rather systemative overestimate of about 0.5 kcal/mol, whereas HF-r$^2$SCAN-DC4 is substantially less for about half the clusters. The 3-body plot shows them being almost identical. But in the total error, we see that HF-r$^2$SCAN-DC4 is far more systematic, as HF-SCAN makes errors of opposite sign, while HF-r$^2$SCAN-DC4 is always an overestimate of about 0.2 kcal/mol. This consistency is important on the plot (d), showing the interaction energy of the 8 hexamers. Because HF-r$^2$SCAN-DC4 is so consistent, it gets the ordering in interaction energies of all clusters correct, whereas HF-SCAN incorrectly predicts that the interaction energy in the bag is higher than that of the chair. The MAE of HF-

r$^2$SCAN-DC4 is 0.19 kcal/mol, <0.22 kcal/mol for HF-SCAN. On average, HF-r$^2$SCAN-DC4 also improves individual $K$-body contributions to the interaction energies, except for $K$ = 4, where both are marginally small (Supplementary Fig. 5). This MBE test shows us that the improvement of HF-r$^2$SCAN-DC4 over HF-SCAN for the water hexamer interaction energies seen also for the relative isomer energies (Fig. 1b) is systematic and does not result from the error cancellations between different $K$-body contributions (for the detailed information of water hexamer isomerization energy in Fig. 1, see Supplementary Fig. 9).

**Water ⋯ cytosine interaction energies**
In Fig. 4, we study the performance of different variations for microhydration of cytosine, by specifically focusing on the interaction energies in water ⋯ cytosine complexes. We generate these complexes as described in Supplementary Note 7, and in all of them, water interacts with cytosine through the hydrogen bond formed between the hydrogen atom in water and the oxygen atom in cytosine. For each complex, the errors of HF-r$^2$SCAN-DC4 are small, and with the MAE of 0.09 kcal/mol, it is the best performer in Fig. 4.

The errors of HF-SCAN are much smaller here than for cytosine dimers (Fig. 1a), in which the role of dispersion is more important. Nevertheless, HF-r$^2$SCAN-DC4 provides here a significant improvement over HF-SCAN. It is also interesting to observe what happens after we add the dispersion correction to HF-r$^2$SCAN and its SC counterpart. In the case of HF-r$^2$SCAN, the errors in the interaction energies are greatly reduced (roughly by a factor of 6 on average). In stark contrast, adding D4 to SC-r$^2$SCAN significantly deteriorates its accuracy, as SC-r$^2$SCAN already overbinds water ⋯ cytosine complexes and D4 makes the overbinding stronger.

**Wide applicability of HF-r$^2$SCAN-DC4**
A functional that works extremely well for pure water but nothing else is not widely applicable. Recently, GMTKN55 of 55 databases has become a popular benchmark for testing the accuracy of density functionals for main-group chemistry. Figure 5 has been designed to illustrate performance of functionals for both pure water and on the GMTKN55 database simultaneously. The water metric ($y$-axis on the left) combines most of the reactions with water used in this paper, and is carefully defined in Supplementary Note 8.

Figure 5a shows errors on GMTKN55 on the $x$-axis and errors on the water metric on the $y$-axis, each in kcal/mol. The $x$-axis ranges from about 3–10 kcal/mol, spanning the performance of modern approximations for main group chemistry, such as atomization energies. The $y$-axis range is much smaller, running <4.0 kcal/mol, reflecting the

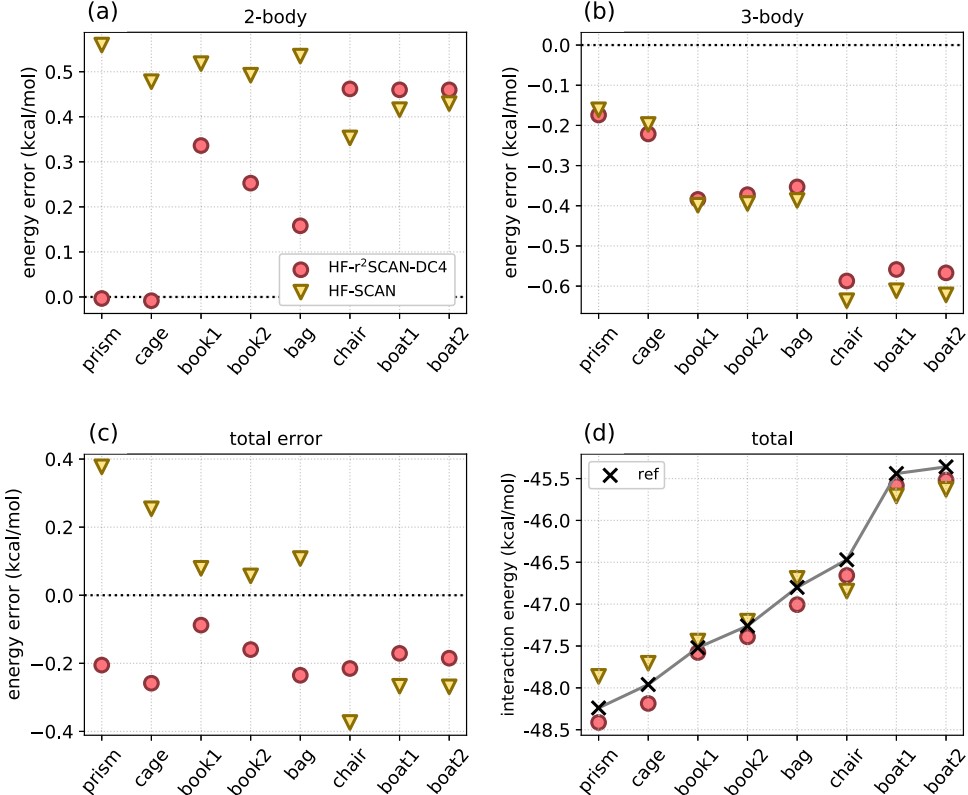

**Fig. 3 | K-body interaction energy errors. a** $K = 2$, **b** $K = 3$, and **c** total, and **d** the interaction energy for 8 water hexamers. (For higher order $K$-body interaction energies, see Supplementary Figs. 5 and 6.) Geometries and CCSD(T)/CBS reference interaction energies are from ref. [13]. The MAEs of HF-r$^2$SCAN-DC4 and HF-SCAN are 0.19 kcal/mol and 0.22 kcal/mol, respectively.

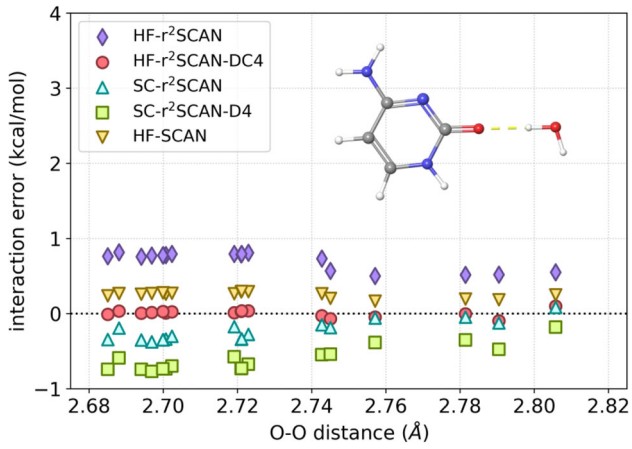

**Fig. 4 | Errors in interaction energies of water ⋯ cytosine complexes.** Atom color code: C, gray; O, red; N, blue; and H, white. Errors are sorted by the distance between the oxygen atom in cytosine and the oxygen atom in water. Reference interaction energies have been computed at the DLPNO-CCSD(T)-F12/AVQZ level of theory.

much smaller magnitude of NCIs in water, and how high accuracy needs to be in order to have an accurate model for water. Here, HF-SCAN sets a high standard, with a water error near 1.0 kcal/mol (the chemical accuracy claimed in ref. [3]), while most standard-use functionals cannot compete. On the other hand, SCAN is designed mainly to improve materials calculations without the cost of a hybrid functional, and HF-SCAN has a high error on GMTKN55 (about 9 kcal/mol). Popular functionals have much smaller GMTKN55 errors, but perform worse on water. We also show the many combinations of HF-r$^2$SCAN-

DC4 that do not include all the right ingredients, showing they all perform less well on water than HF-SCAN. We finally include $\omega$B97M-V functional[67], which might be considered the DFT gold-standard here, with the smallest errors for both water and main-group chemistry. But this range-separated functional with nonlocal correlation functional is far more expensive to compute than most functionals, including its own D4 variant,[68] and is less practical for DFT-MD simulations than e.g., SCAN. We have included it here only to show what is possible in principle with DFT.

But the performance of HF-r$^2$SCAN-DC4 is remarkable. Its errors on both water and the GMTKN55 dataset are almost half of those of HF-SCAN. No other functional in our collection comes close for water. Clearly, all the chemically-inclined approximations which are comparable for main-group chemistry do much worse.

In Fig. 5b, we show the hexagon plots comparing the MAEs of several density functionals, where the position of five vertices denote the MAEs for individual water-based datasets, while the sixth vertex denotes the overall performance of the functionals for the whole GMTKN55 databases, as measured by the weighted-mean-absolute-deviation-2 (WTMAD-2). It is the MAE for all the reactions from these five water-based datasets that we use as the quantity on the y-axis in Fig. 5a. The size of the hexagon of HF-r$^2$SCAN-DC4 is the closest to that of more costly $\omega$B97M-V. We can also see that the performance of HF-r$^2$SCAN-DC4 is far superior to that of HF-SCAN. M062X-D3(0), a meta-hybrid that is very accurate for small organic molecules,[40] and yields WTMAD-2 which is slightly lower than that of HF-r$^2$SCAN-DC4. But, for water simulations, M062X-D3(0) is nowhere close to HF-r$^2$SCAN-DC4, as can be seen from the position of the remaining five vertices.

## Discussion

In refs. [31,65], we proposed DC(HF)-DFT, a DC-DFT procedure that discriminately uses HF densities based on the density sensitivity criterion.

(a)

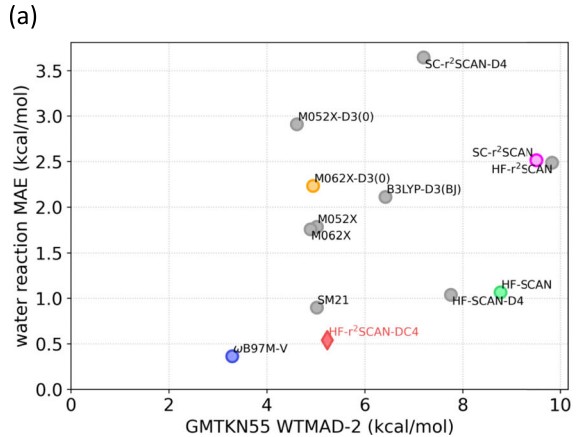

(b)

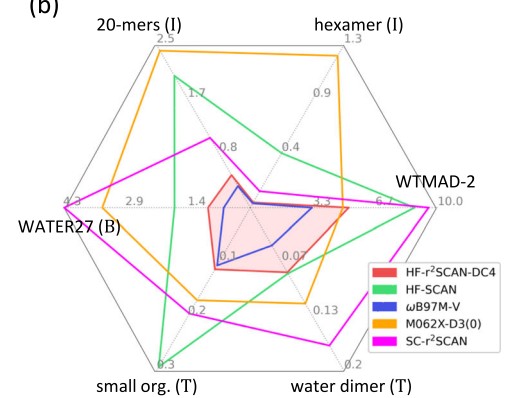

**Fig. 5 | Performance of HF-r²SCAN-DC4 and other conventional functionals.**
**a** The mean absolute error (MAE) for the water-based reactions appear in this work (hexamer isomer energies, water 20-mers isomer energies, WATER27 binding energies, water-small organic molecule interaction energies, and water dimer interaction energies) versus the weighted-mean-absolute-deviation-2 (WTMAD-2) for the GMTKN55 database for selected functionals. For a further description of the reactions used in the *y*-axis, see Supplementary Note 8. HF-SCAN-D4 functional

used here is from ref. [74]. **b** The hexagon plot with MAEs for selected water-based datasets and WTMAD-2 values for the whole GMTKN55 databases (for WTMAD-2 values for other GMTKN55 database, see Supplementary Fig. 10). Abbreviations of isomerization (I), binding (B), and interaction (T) energy are noted in the vertex caption. MAEs of HF-r²SCAN-DC4 for individual GMTKN55 datasets are shown in Supplementary Table 1. In Supplementary Fig. 11, we give further details about the interaction energies used in the water-small organic molecule dataset.

The main idea of DC(HF)-DFT is to use HF-DFT for density-sensitive (DS) reactions and SC-DFT for density-insensitive (DI) reactions (possible spin-contaminations of the HF results are also taken into account as detailed in Supplementary Note 3). While we consider DC(HF)-DFT a state-of-the-art DC-DFT procedure, for our HF-r²SCAN-DC4 we use HF-DFT, meaning that the functional is always evaluated on the HF density regardless of the sensitivity criterion. To use DC(HF)-DFT, we need to compute density sensitivity for each reaction of interest and possibly make adjustments to its cutoff value which is used to declare whether a given reaction is DS or DI.[31,65] This would also require having two sets of D4 parameters, one for DS and the other for DI reactions. All these efforts would undermine the ease of use of r²SCAN, which is a general-purpose functional. For this reason and encouraged by the very good performance of HF-DFT with SCAN-like functionals[32,42], we employ HF-DFT[69] as a DC-DFT procedure for HF-r²SCAN-DC4. While our HF-r²SCAN-DC4 can be routinely used by applying it to HF orbitals without ever needing to calculate density sensitivity of a given reaction, the use of DC-DFT principles and density sensitivity is vital for our training of HF-r²SCAN-DC4 as explained above.

To illustrate what can happen when these principles are not applied, we show results from ref. [64]. This is a version of HF-r²SCAN-D4, but where all reactions in GMTKN55 were used, and the WTMAD-2[40] was used as the cost function instead. Figure 6 illustrates the results for the larger water clusters. In every case, they are noticeably worse than ours. Moreover, (d) shows that, apart from matching on WTMAD-2 measure, HF-r²SCAN-DC4 yields more accurate results in every other case.

The work of ref. [3] was a breakthrough in models for water, showing that, by using the principles of DC-DFT, a moderate-cost density functional approximation approached chemical accuracy for many relevant properties of small water clusters. However that functional is lacking in dispersion corrections, yielding large errors for energetics between organic and biological molecules. It also inherits some of the numerical issues of the original SCAN functional, which have been eliminated by using r²SCAN instead in most other applications. However, the small differences between these two wreak havoc on the much smaller scale of subtle energy differences of water clusters.

The present work shows that, by a very careful application of the principles of DC-DFT, all these difficulties can be overcome, and even greater accuracy achieved for pure water, while still including

dispersion for other molecules where it can be vital. Finding the correct parameters depends crucially on training on only DI chemical reactions, as inclusion of DS reactions yields suboptimal values for the parameters.

Even if HF-r²SCAN-DC4 could be run at close to meta-GGA cost, KS-DFT MD simulations are typically far more costly than MD with machine learning (ML) interatomic potentials. But accurate force-field generation requires highly accurate reference energetics data as a training set, and CCSD(T) or Quantum Monte Carlo (QMC) methods are frequently used as reference methods these days.[12] Due to the large computational cost for such ab initio calculation, a more practical yet accurate method is in demand, and HF-r²SCAN-DC4 can replace them for calculating moderately large biomolecular systems. We suggest HF-r²SCAN-DC4 be tested and applied in solution wherever practical.

## Methods

The basic principles of DC-DFT are covered elsewhere in the literature[19,22], and reviewed in the supplementary information. In most KS-DFT calculations, the error in the density has a negligible effect on the energy errors. But sometimes the error in a SC density leads to a noticeable contribution, which can be reduced if a more accurate density is used instead. For many semilocal exchange-correlation approximations in molecular calculations, when a calculation is density sensitive, often the HF density then yields significantly smaller energy errors. These principles have led to improved energetics in reaction barrier heights, electron affinities, and also for the ground state geometries of noncovalent interaction systems, etc.[20,70–72]

Application of the principles of DC-DFT is subtle in the case of r²SCAN-D4, because of the need to separate out the error due to density correction from the fitting of the D4 corrections. For example, for halogen bonds, the density-driven errors are far larger than dispersion corrections, so all fitting must be done on density-corrected energetics. Moreover, when empirical functionals contain parameters, such parameters should be fit only on density-insensitive calculations, so that the parameters optimize the true functional error.

With these principles in mind, we find the parameters for HF-r²SCAN-DC4 using the density-insensitive calculations in the GMTKN55 dataset as a training set while using water ⋯ water pair interaction energy as a validation set. We find their optimum values by minimizing MAE values over all such cases. This is detailed in Supplementary Note 4. This is why we use the acronym DC4 instead of D4, meaning

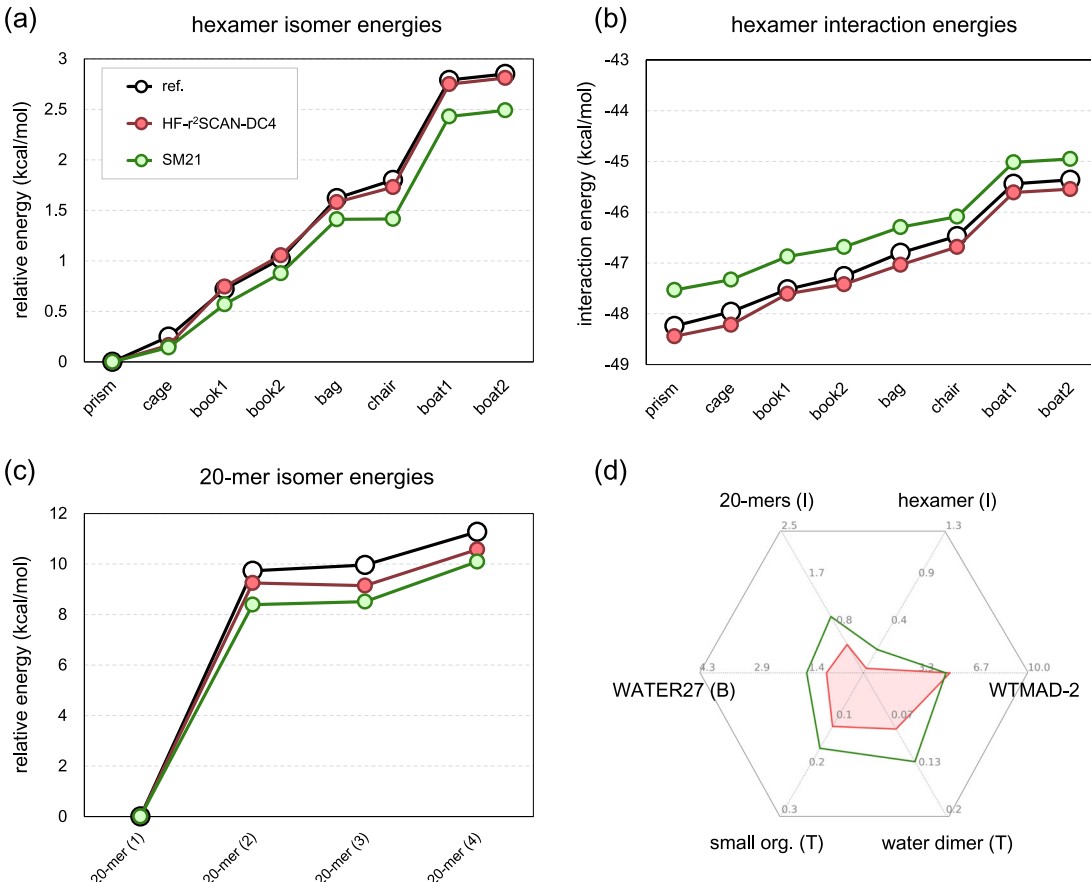

**Fig. 6 | Comparison between HF-r²SCAN-DC4 and SM21. a** Water hexamer isomerization energy, **b** water hexamer interaction energy, **c** water 20-mer isomerization energy, and **d** hexagonal plot same as Fig. 5b. SM21 (green) is HF-r²SCAN but with different D4 parameters obtained in ref. [64].

that we use the principles of DC-DFT to find the underlying D4 parameters.

## Data availability

Individual reaction energy and density sensitivity values of HF-r²SCAN for GMTKN55 reported in this study are available at http://tccl.yonsei.ac.kr and also provided in the Supplementary Data file. DLPNO-CCSD(T)-F12/TightPNO interaction energies and cartesian coordinates of cytosine ⋯ water, aspirin ⋯ water, and water ⋯ water dimers are provided in the Supplementary Data file.

## Code availability

A Pyscf script for HF-r²SCAN-DC4 is available at http://tccl.yonsei.ac.kr.

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

## Acknowledgements

E.S., S.S., Y.K., and H.Y. are grateful for support from the National Research Foundation of Korea (NRF-2020R1A2C2007468 and NRF-2020R1A4A1017737) and Samsung Electronics (IO211126-09176-01). K.B. acknowledges funding from NSF (CHE-2154371). S.V. acknowledges funding from the Marie Skłodowska-Curie grant 101033630 (EU's Horizon 2020 programme). We thank John Perdew and Francesco Paesani and his group for many useful discussions.

## Author contributions

S.S., Y.K., and H.Y. performed the calculation, S.S., Y.K., S.V., H.Y., E.S., and K.B. analysed the result, E.S. and K.B. supervised the work.

## Competing interests

The authors declare no competing interests.
