## [Peer Review File · Nature Communications]

REVIEWER COMMENTS

Reviewer #1 (Remarks to the Author):

I enjoyed reading this manuscript, which is well written and easy to follow. Agreeing with the authors, I think this work is a major extension of density functional theory, enabling its applications to bio-related systems with chemical accuracy. The key improvement over the quality of the functional is its balanced consideration between density correction and dispersion correction. I have following comments for the authors to consider.

Major points:

1. In the subsection of "HF-r2SCAN-DC4, an integratively designed DC-DFT procedure", the authors outlined a few "key features" of the new functional. It seems to me that these features are indeed outcomes, not procedures per se. Instead, I think the authors outlined the new features in "III. Methods", where they said "to separate out the error due to density correction from the fitting of the D4 corrections" and "parameters should be fit only on density-insensitive calculations".

2. The new functional uses the HF density, so no self-consistent density is made available or at least required. What I was trying to say here is that this work focuses only on the energetics. What about electronic properties, whose calculation depends on orbitals? If the SC version gives worse results than the HF version for some systems, as the authors discussed in the main text, does this mean that properties computed by this new functional are not necessarily better than the original?

Minor points:

- * Abstract, "which recovers", missing "not only" between the two words;
- * First paragraph in the "The importance of the functional" subsection, B3LYP requires a citation;
- * Figure 1, legends in (a) are shared by all others, so they must be moved outside (a) to reflect the fact that all symbols in the Figure represent the same meanings. Also, (d) has 4 curves. What is its extra curve?
- * Page 3, left column, "Perdew and co-workers have developed r2SCAN to address these issues of SCAN", for the purpose of better readership, it is helpful to use a few words to summarize its key feature, i.e, why "r2"?
- * Page 7, left column, "This is why we use the acronym DC4 instead of D4, meaning D4 accounts for density correction", should the second "D4" be "DC4"?
- * Page 7, right column, "one for DS and one for DI reactions", the second "one" should be "the other".

Reviewer #2 (Remarks to the Author):

Extending density functional theory with near chemical accuracy beyond pure water

Noteworthy results

This article builds earlier on results by others that showed that the Density Functional Theory (DFT) based on the so-called SCAN approximation can yield chemical accuracy for the phases of pure water, especially when employed in tandem with Hatree-Fock (HF) density corrected DFT, i.e., HF-DFT. More recent work has focused on examining the role of dispersion forces, which are very relevant for biomolecular systems. The present work introduces a pragmatic computational scheme called HF-r2SCAN-DC4, which adds carefully parameterized dispersion correction (DC4) to the HF-r2SCAN scheme of Perdew and co-workers. The results of the present work amply demonstrate the superiority of HF-r2SCAN-DC4 compared to other competitive models.

Significance to the field and related fields

Potentially, important work that is of the caliber that deserves to be published in Nature

Communications.

Does the work support the conclusions and claims?

While the results contained in the article convincingly demonstrate that the HF-r2SCAN-DC4 scheme is worthy of being employed to investigate more complex biomolecular systems, However, to do so, will require the application of AI/ML methodologies to bridge to physical systems too large to be handled by purely ab initio methodologies. Some comments on the feasibility of HF-r2SCAN-DC4 in the domain of biomolecular assemblies is warranted.

Reviewer #3 (Remarks to the Author):

**Extending density functional theory with near chemical accuracy beyond pure water
by S. Song, S. Vuckovic, Y. Kim, E. Sim, and K. Burke**

In this manuscript, Burke and collaborators put forward a new ab initio computational method to model the electronic degrees of freedom in water and a range of solutes. Specifically they propose a density functional that carries the somewhat unwieldy name of HF-r²SCAN-DC4. The reasons for the nomenclature are given in the text, along with a description of the actual conceptual contents of the various modifications and implementations of the Kohn-Sham framework. In brief, the functional is based on SCAN, and the authors' work lies in combining successfully density and dispersion corrections. Density errors are a general source of problems in DFT. Accounting correctly for dispersion is inherently challenging due to the long-range nature of van der Waals forces.

The paper contains a systematic and, as far as I can tell, very careful examination of the performance of the new theory. A wide range of relevant problems in water is considered, from solvation of aspirin to small clusters of water molecules and much more. The authors conclude their paper by suggesting that their new theory "be tested and applied in solution wherever practical". There is high interest and relevance of the considered problem. It goes with no arguing that computationally accurate models of water are important in many scientific fields.

I enjoyed reading the paper. I think in particular that the SM is very clear and helpful for more general readers. I summarize below some points that the authors might want to consider in a revision of their manuscript. In my view the manuscript presents a nontrivial step up in accuracy over the already advanced state of the art in the field. I have no doubts that the paper will have significant impact and I hence recommend publication in Nature Communications.

1) Some more (brief) background in the SM on the details of the underlying approaches would help. In particular the section S3, I think, could be extended.

2) Concerning Fig.3, the authors write "whereas HF-SCAN incorrectly predicts that the interaction energy in the chair is higher than that of the boat." Maybe I am misunderstanding, but should that be: chair is lower than the boat? Please clarify.

3) The authors show results from as they call it "goldstandard" omegaB97M-V functional [66] and say that this is far more expensive to compute than most functionals. Can this be made more precise, at least in orders of magnitude computational cost/runtime/storage requirements? I do not mean to include a thorough technical account, but to give an illustration. This point also applies to a comparison with the other approaches. So some more information on the actual level of computational cost would be helpful, I think.

4) Commenting on the feasibility of the new theory within a dynamical setting would be

worthwhile.

Responses to Reviewer #1's comments

I enjoyed reading this manuscript, which is well written and easy to follow. Agreeing with the authors, I think this work is a major extension of density functional theory, enabling its applications to bio-related systems with chemical accuracy. The key improvement over the quality of the functional is its balanced consideration between density correction and dispersion correction. I have following comments for the authors to consider.

Response: We thank the reviewer for carefully reading the manuscript and appreciating its significance.

Comment 1.1: *In the subsection of "HF-r2SCAN-DC4, an integratively designed DC-DFT procedure", the authors outlined a few "key features" of the new functional. It seems to me that these features are indeed outcomes, not procedures per se. Instead, I think the authors outlined the new features in "III. Methods", where they said "to separate out the error due to density correction from the fitting of the D4 corrections" and "parameters should be fit only on density-insensitive calculations".*

Response: "Features" is the wrong word. We change the sentence. [C4]

Comment 1.2: *The new functional uses the HF density, so no self-consistent density is made available or at least required. What I was trying to say here is that this work focuses only on the energetics. What about electronic properties, whose calculation depends on orbitals? If the SC version gives worse results than the HF version for some systems, as the authors discussed in the main text, does this mean that properties computed by this new functional are not necessarily better than the original?*

Response: In principles, $n(r) = \delta E / \delta v(r)$ yields the correct density for HF-DFT, which differ from either SC or HF densities. While finite energies and their differences are insensitive to the difference, response properties are. A simple example is the dipole moment, which can be found either from energy in response to an external field, or as an integral over density. The simplest way in HF-DFT is energy with regard to the field, and we do not know if HF-DFT does better than SC-DFT for this or any other response property.

Comment 1.3: *Abstract, "which recovers", missing "not only" between the two words;*

Response: We added “not only” to the sentence. [C1]

Comment 1.4: *First paragraph in the "The importance of the functional" subsection, B3LYP requires a citation;*

Response: The citations are added. [C2]

Comment 1.5: *Figure 1, legends in (a) are shared by all others, so they must be moved outside (a) to reflect the fact that all symbols in the Figure represent the same meanings. Also, (d) has 4 curves. What is its extra curve?*

Response: A green square-marked plot in (d) is SC-r²SCAN-D4 whose label was in (c). We separate the Figure 1 legend for clarity.

Comment 1.6: *Page 3, left column, "Perdew and co-workers have developed r2SCAN to address these issues of SCAN", for the purpose of better readership, it is helpful to use a few words to summarize its key feature, i.e, why "r2"?*

Response: We agree with the reviewer, and we explain what r²SCAN is. [C3]

Comment 1.7: *Page 7, left column, "This is why we use the acronym DC4 instead of D4, meaning D4 accounts for density correction", should the second "D4" be "DC4"?*

Response: Change is made to clarify this. [C8]

Comment 1.8: *Page 7, right column, "one for DS and one for DI reactions", the second "one" should be "the other".*

Response: The reviewer is right. We changed “one” to “the other”. [C9]

”

Responses to Reviewer #2's comments

This article builds earlier on results by others that showed that the Density Functional Theory (DFT) based on the so-called SCAN approximation can yield chemical accuracy for the phases of pure water, especially when employed in tandem with Hatree-Fock (HF) density corrected DFT, i.e., HF-DFT. More recent work has focused on examining the role of dispersion forces, which are very relevant for biomolecular systems. The present work introduces a pragmatic computational scheme called HF-r2SCAN-DC4, which adds carefully parameterized dispersion correction (DC4) to the HF-r2SCAN scheme of Perdew and co-workers. The results of the present work amply demonstrate the superiority of HF-r2SCAN-DC4 compared to other competitive models.

Response: We thank the referee for the positive evaluation of the originality of our research.

Comment 2.1: *While the results contained in the article convincingly demonstrate that the HF-r2SCAN-DC4 scheme is worthy of being employed to investigate more complex biomolecular systems, However, to do so, will require the application of AI/ML methodologies to bridge to physical systems too large to be handled by purely ab initio methodologies. Some comments on the feasibility of HF-r2SCAN-DC4 in the domain of biomolecular assemblies is warranted.*

Response: We added a paragraph to the paper. [C10]

Responses to Reviewer #3's comments

The paper contains a systematic and, as far as I can tell, very careful examination of the performance of the new theory. A wide range of relevant problems in water is considered, from solvation of aspirin to small clusters of water molecules and much more. The authors conclude their paper by suggesting that their new theory "be tested and applied in solution wherever practical". There is high interest and relevance of the considered problem. It goes with no arguing that computationally accurate models of water are important in many scientific fields. I enjoyed reading the paper. I think in particular that the SM is very clear and helpful for more general readers. I summarize below some points that the authors might want to consider in a revision of their manuscript. In my view the manuscript presents a nontrivial step up in accuracy over the already advanced state of the art in the field. I have no doubts that the paper will have significant impact and I hence recommend publication in Nature Communications.

Response: We thank the reviewer for carefully reading the manuscript and appreciating its significance.

Comment 3.1: *Some more (brief) background in the SM on the details of the underlying approaches would help. In particular the section S3, I think, could be extended.*

Response: We added a detailed explanation about (DC)HF-DFT in section S3. [SC1]

Comment 3.2: *Concerning Fig.3, the authors write "whereas HF-SCAN incorrectly predicts that the interaction energy in the chair is higher than that of the boat." Maybe I am misunderstanding, but should that be: chair is lower than the bag? Please clarify.*

Response: The reviewer is right. We are very grateful and deeply sorry for finding the typo. [C5]

Comment 3.3: *The authors show results from as they call it "goldstandard" omegaB97M-V functional [66] and say that this is far more expensive to compute than most functionals. Can this be made more precise, at least in orders of magnitude computational cost/runtime/storage requirements? I do not mean to include a thorough technical account, but to give an illustration. This point also applies to a comparison with the other approaches. So some more information on the actual level of computational cost would be helpful, I think.*

Response: HF-r2SCAN part and ω B97M part are quite the same for the computational cost since the electron-repulsion-integral generation is the bottleneck for both calculations. However, for DC4 and VV10 it is significantly different. The computational cost for DC4 is negligibly small compared to the non-local density functional VV10. In paper [J. Chem. Theory Comput. 2018, 14, 5725–5738], Najibi et al., introduced ω B97M-D3(BJ) functional which uses D3(BJ), a former version of DC4, instead of VV10 for the “faster variants with similar accuracy”. [C6]

Comment 3.4: *Commenting on the feasibility of the new theory within a dynamical setting would be worthwhile.*

Response: HF-SCAN has been used in tandem with MD to predict the condensation of water and the density of water as a function of temperature. Our HF-r2SCAN-DC4 can be used within the same MD setting, and we believe that it would give further improvements over HF-SCAN for solutions. Running ab into MD simulations of solutions that are based on HF-r2SCAN-DC4 will be the objective of our future work and we have added a paragraph. [C10]

Detailed List of Changes

[C1] “Not only” is added in the following sentence.

“Systematic application of the principles of density-corrected DFT yields a functional (HF-r²SCAN-DC4) which recovers and not only improves over HF-SCAN for pure water, but also captures vital non-covalent interactions in biomolecules, making it suitable for simulations of solutions.”

[C2] The relevant papers are cited. Reference 33~36.

33. Becke, A. D. Density-functional exchange-energy approximation with correct asymptotic behavior. *Phys. Rev. A* 38, 3098–3100 (1988).

34. Lee, C., Yang, W. & Parr, R. G. Development of the colle-salvetti correlation-energy formula into a functional of the electron density. *Phys. Rev. B* 37, 785–789 (1988).

35. Becke, A. D. Density-functional thermochemistry. iii. the role of exact exchange. *The J. Chem. Phys.* 98, 5648–5652 (1993).

36. Stephens, P. J., Devlin, F. J., Chabalowski, C. F. & Frisch, M. J. Ab initio calculation of vibrational absorption and circular dichroism spectra using density functional force fields. *The J. Phys. Chem.* 98, 11623–11627, (1994)

[C3] Modified the sentence by adding the r²SCAN functional description.

Previous sentence: “Perdew and co-workers have developed r²SCAN to address these issues of SCAN,[42] but as we show below, a standalone version of HF-r²SCAN is much less accurate for water simulations than HF-SCAN.”

Changed to: “To address these issues of SCAN, Perdew and co-workers developed the regularized-restored SCAN functional (r²SCAN), which regularizes SCAN but restores SCAN’s adherence to exact constraints.[42] But, as we show below, a standalone version of HF-r²SCAN is much less accurate for water simulations than HF-SCAN.”

[C4] Modified the sentence by changing “features” to “results”.

Previous sentence: “This yield HF-r²SCAN-DC4, which has the following key features.”

Changed to: “This yield HF-r²SCAN-DC4, which produces the following key results.”

[C5] A typo corrected.

Previous sentence: “whereas HF-SCAN incorrectly predicts that the interaction energy in the chair is higher than that of the boat.”

Changes to: “whereas HF-SCAN incorrectly predicts that the interaction energy in the bag is higher than that of the chair.”

[C6] A phrase added to imply a larger computational cost for calculating VV10.

[C7] Added a brief description of the validation set.

Previous sentence: “With these principles in mind, we find the parameters for HF-r2SCAN-DC4 using the density-insensitive calculations in the GMTKN55 dataset.”

Changes to: “With these principles in mind, we find the parameters for HF-r2SCAN-DC4 using the density-insensitive calculations in the GMTKN55 dataset as a training set while using water···water pair interaction energy as a validation set.”

[C8] Changed the sentence for clarity.

Previous sentence: “This is why we use the acronym DC4 instead of D4, meaning D4 accounts for density correction.”

Changes to: “This is why we use the acronym DC4 instead of D4, meaning that we use the principles of DC-DFT to find the underlying D4 parameters.”

[C9] Wording changed for clarity.

Previous sentence: “This would also require having two sets of D4 parameters, one for DS and one for DI reactions.”

Changes to: “This would also require having two sets of D4 parameters, one for DS and the other for DI reactions.”

[C10] A paragraph that describes the application with HF-r²SCAN-DC4 is added.

“Even if HF-r²SCAN-DC4 could be run at close to meta-GGA cost, KS-DFT MD simulations are typically far more costly than MD with machine learning (ML) interatomic potentials. But accurate force-field generation requires highly accurate reference energetics data as a training set, and CCSD(T) or Quantum Monte Carlo methods are frequently used as reference methods these days.[12] Due to the large computational cost for such *ab initio* calculation, a more practical yet accurate method is in demand, and HF-r²SCAN-DC4 can replace them for calculating moderately large biomolecular systems.”

[SC1] A detailed explanation about DC(HF)-DFT is added.

“Since HF-DFT uses the HF density as a proxy for the exact density, we only use it when there is little or no spin contamination. We calculate the expectation values of the spin-squared operator, S^2 , and only use the HF density if the $\langle S^2 \rangle$ from the HF calculation deviates less than

10% from the exact $\langle S^2 \rangle$ as discussed in Refs. [13] and [15]. Otherwise, we use the self-consistent density.”

[SC2] Please note that the DC4 parameter set has been slightly modified from $(s_8, a_1, a_2)=(-0.33, 0.38, 4.64)$ to $(-0.36, 0.23, 5.23)$. During the revision, we found that when GMTKN55 is used as the training set, the choice of the optimal parameter set can change depending on the technical details of the computational setup such as DFT grid information, two-electron operator fitting scheme, etc. To make the selection more rigorous, (density-insensitive) water-water pair interaction energies were used as a validation set. The parameter change shifted the MAE for GMTKN55 density insensitive (all) cases from 1.217 (2.412) to 1.215 (2.419) kcal/mol. All data in the manuscript has negligible changes that are barely visible to the naked eye. We mention this in the manuscript (on page 9, [C7]) and also add a detailed explanation in the supporting information section S4 (on page S2).

“However, the density-insensitive reactions in GMTKN55 largely fall into two distinct parameter groups for HF- r^2 SCAN: s_8 has a negative value for non-covalent interactions, but is positive for the rest. The difference in MAE of density-insensitive cases between those two groups is miniscule (below 0.01 kcal/mol). For example, $(s_8, a_1, a_2)=(-0.20, 0.07, 6.50)$ gives 1.209 kcal/mol for the density-insensitive MAE while $(0.39, 0.09, 7.02)$ gives 1.210 kcal/mol. Such a difference is not meaningful. Small changes in computational details such as DFT grid information, two-electron operator fitting scheme, etc. changes the values of the parameters, since reaction energy errors and density-sensitivity values can be changed by 0.01 kcal/mol with those changes. To eliminate this ambiguity while ensuring accuracy in water interactions, we include the density-insensitive water···water pair interaction energy as a validation set. The two most stable water hexamers, the prism and the cage, are used to calculate the water···water 2-body interaction energy error per dimer, relative to CCSD(T)/CBS in Ref. [20]. We multiply its weight by 7 in our loss function to produce a better defined minimum and regularize the result (if we used 1, it has no effect; if we used 1000, we simply fit to this data). We can rationalize this value by noting that the mean density-sensitivity of these pairs is 0.27 kcal/mol, which is about 1/7th of our density-sensitivity threshold. The resulting values for the three parameters are: -0.36, 0.23, 5.23 for s_8 , a_1 , and a_2 each.”

REVIEWERS' COMMENTS

Reviewer #3 (Remarks to the Author):

In their revision the authors have addressed the points in my original report in a thorough way. I hence recommend publication of this manuscript in Nature Communications.